# An Adversarial Attack Framework via Decision Boundary Drift for Continual Learning

## Abstract

Continual learning (CL) is widely used in open environments due to its dynamic adaptive ability. However, our further analysis reveals that due to the weight drift phenomenon occurred in the parameters update stage, the model is under serious risk of adversarial attacks. To tackle this issue, we propose an Adversarial Attack framework based on Decision Boundary drift (AADB). It includes: (1) an adversarial sample generation method based on the decision boundary drift phenomenon, which significantly reduces the model's classification accuracy to 4.41%; (2) a composite loss function based on similarity loss and adversarial loss to optimize adversarial samples, which can reduce the classification accuracy of adversarial samples without significantly affecting the quality of adversarial samples; (3) an adversarial sample attack method that distorts the decision boundary of model by mixing adversarial samples and normal samples, affecting the model performance; (4) a defense framework based on dynamic feature consistency, cross-category comparison learning and resilient rejection mechanism, which can suppress the deformation of decision boundary caused by adversarial perturbation and improve the rejection rate of adversarial samples to 40.16%. Experiments on CIFAR-100, Mini-ImageNet, and other datasets prove that the adversarial attack framework has significant effects and provide an algorithmic foundation for the subsequent exploration of the field of continual learning lightweight defense and adaptive attack detection mechanism.

## 1 Introduction

Continual learning (CL) (Van de Ven & Tolias (2019); Wang et al. (2024)) utilizes an incremental knowledge integration mechanism to enable model to adapt to new task distributions and maintain historical task performance. The methods have been widely applied in multiple security-sensitive scenarios, such as preserving autonomous driving (Verwimp et al. (2023); Yang et al. (2025)), wise information technology of medicine (Perkonigg et al. (2021); Li & Jha (2024); Wang et al. (2024)), etc. Although significant progress has been made in mitigating catastrophic forgetting in continual learning, existing research lacks in-depth exploration of the robustness of continual learning. To this end, researchers have proposed various attack strategies against continual learning frameworks (Abbasi et al. (2024); Li & Ditzler (2022; 2023); Kang et al. (2023)). The most widely used method among them is to contaminate task data to interfere with the performance of model on current or historical tasks, which typically rely on injecting misleading samples or perturbing data distributions to induce decision biases in the model.

With the continuous evolution of CL algorithms, the model exhibits stronger feature extraction capabilities and optimizes the decision boundaries of categories, thereby demonstrating high robustness to traditional noise disturbances. Therefore, adversarial samples generated based on noise have poor semantic consistency and visual concealment, and are easily recognized and filtered out by anomaly detection mechanisms. In addition, we find that the CL parameters will suffer from irreversible drift when updating, reducing the robustness of model.

In response to these shortcomings, this study validates the phenomenon of decision boundary drift and used it as a novel perspective for adversarial attacks. The article proposes an Adversarial Attack framework based on Decision Boundary drift (AADB), which is the first method to systematically utilize the dynamic evolution characteristics of decision boundaries in continual learning to con-

struct adversarial samples. AADB simulates the category decision boundary drift behavior caused by continual learning, accurately capturing the changing trajectories of decision boundaries at different training stages, and thus constructing adversarial samples with semantic consistency, visual concealment, and higher attack effectiveness. Figure 2 shows the overall architecture of the algorithm proposed in this study.

The core component of this framework is the adaptive attack module, which constructs targeted perturbations based on dynamic drift of category centers and introduces task decay factors to constrain the strength of decision boundary drift, thereby achieving effective adversarial attacks while avoiding overall performance collapse caused by excessive perturbations in the model. Unlike previous approaches that focus only on attack success rate, the framework enhances both the representation ambiguity of feature space and the transferability of decision boundaries. In addition, we have designed a defense module as a supplementary component to framework, which achieves real-time detection and filtering of adversarial samples by constructing a multi-level cascaded active defense system that integrates resilient rejection mechanism, local feature consistency, and global uncertainty.

We conducted experiments on multiple mainstream classification models, and AADB achieved significant and stable attack effects under different model architectures. Even without further iteration optimization, its initial adversarial samples can reduce the classification accuracy of target model by approximately 20% on average. On the CIFAR-100 dataset, our attack method successfully significantly reduced the backward migration metric (Lopez-Paz & Ranzato (2017)) of the AFEC algorithm (Wang et al. (2021)) from the baseline value of -2.9% to -40.38%. In addition, the above defense mechanism can effectively intercept 40.16% of adversarial samples.Our main contributions are:

- We have demonstrated the phenomenon of decision boundary drift in continual learning and its potential impact on model robustness.
- We propose an adversarial attack framework based on decision boundary drift (AADB), which includes high-dimensional feature level adversarial sample generation for continual learning, adversarial attack methods, and multi-level cascaded active defense mechanisms.
- We have validated the effectiveness of AADB on multiple models and CL algorithms. Experiments have shown that this attack method significantly reduces the performance of model.

## 2 BACKGROUND AND MOTIVATION

### 2.1 PRELIMINARIES

**Continual Learning:** Continual learning addresses the scenario where a model sequentially encounters tasks with non-identical data distributions, overcoming the limitations of the traditional i.i.d. assumption. Formally, the CL objective is to find parameters $\theta^*$ that minimize the overall expected risk across all tasks $T_1, T_2, \ldots, T_N$. To address the core issue of catastrophic forgetting, CL adopts the following methods: (i) imposing constraints on parameter updates to protect critical subspaces related to previous tasks, and (ii) memory replay, which approximates past distributions via stored or generated samples to enable explicit rehearsal.

**Adversarial Examples:** Despite their strong performance, deep neural networks are vulnerable to adversarial examples. Adversarial attack reveals the complexity and vulnerability of the decision boundary of model, whose essence is to push the high-dimensional features of samples away from the decision boundary. Moreover, unlike most previous adversarial example generation methods that generate adversarial samples in pixel space, the method in this study generates adversarial samples in high-dimensional feature space. Therefore, in addition to evaluating the attack success rate of adversarial samples, this study also needs to evaluate the quality of adversarial samples.

**Decision Boundary Drift:** Weight drift (Liu & Chang (2025b); Goswami et al. (2024); Toldo & Ozay (2022)) essentially stems from the dynamic accumulation of catastrophic forgetting during the iteration process, which occurs in different components of the model. Decision boundary drift includes both overall drift and local deformation of the decision boundary, which is more difficult to restrict compared to other types of weight drift. To quantify this phenomenon, this study indirectly measures the degree of decision boundary drift by analyzing changes in the feature centers of each

category in the CIFAR-100 dataset. Fig. 1(a) shows that the 10 category centers of the initial task show significant shifts in subsequent tasks, confirming the existence of decision boundary drift. It is worth noting that although each category experiences drift, the drift range is still constrained within a certain area due to the protection mechanism of the continuous learning algorithm for old knowledge. Additionally, it is also evident in Fig. 1(b) that during the CL parameter update process, not only will the overall decision boundary shift, but the local decision boundary will also undergo drastic distortion.

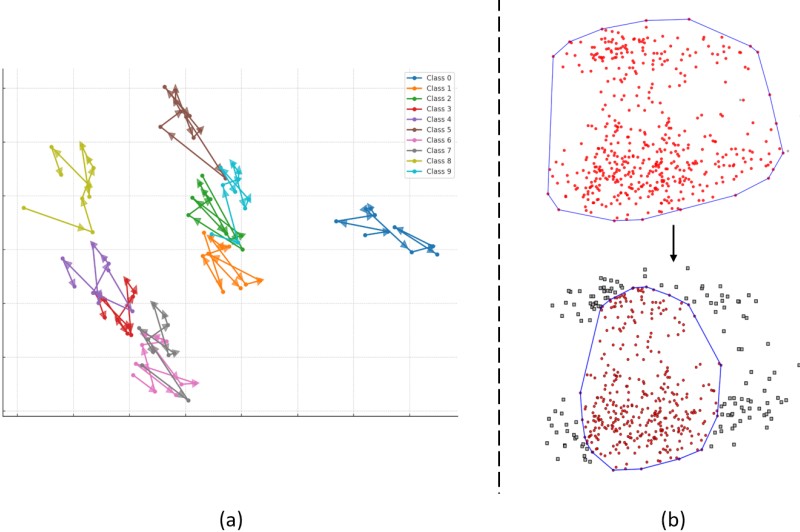

(a)                                                        (b)

Figure 1: (a) Trajectory of class centers for the first subtask over subsequent tasks. (b) the change of decision boundary after learning new task.

## 2.2 THREAT MODEL

**SetUp:** A model learns a sequence of tasks, each characterized by a distinct data distribution with dataset $D_t = (x_i, y_i)_{i=1}^{N_t}$ and class center $C_t$, $t \in \{1, 2, \ldots, T\}$. The fundamental challenge in this setting stems from the non-stationary data distributions across tasks, requiring model to continuously integrate new knowledge while mitigating catastrophic forgetting of previously learned information.

**Attacker's Goals:** The attacker aims to perturb or even permanently distort the decision boundaries learned for old tasks, significantly degrading its performance on previously learned tasks. Moreover, we hope that the added perturbations should be imperceptible to humans and the designed adversarial attack algorithm should have good generality, enabling it to be effectively transferred to different CL models and various task sequence settings.

**Attacker's Capabilities:** To achieve the aforementioned goals, we assume the attacker possesses one or more of the following capabilities:

- The attacker has at least partial white-box or grey-box access to the model, enabling them to obtain intermediate layer output, gradients, classification confidence scores, feature representations, etc.
- The attacker can inject poisoned samples into the data stream of new tasks, thereby influencing the model's iterative optimization process by manipulating the training data.
- The attacker knows how the task sequence is partitioned and the task model is currently learning.
- The attacker possesses sufficient computational resources to generate adversarial examples and can query the model multiple times to probe its responses, thereby optimizing the attack strategy.

## 3 METHODOLOGY

### 3.1 INSIGHTS

The core innovation of the adversarial attack algorithm proposed in this study lies in its deliberate exploitation of decision boundary drift to systematically trigger distortions in the high-dimensional

feature space. Based on the phenomenon of decision boundary drift and artificially amplifying its impact, our approach strategically perturbs individual samples, pushing them away from their original category centers until they cross decision boundaries, ultimately resulting in misclassification. By incorporating both normal and adversarial samples into the fine-tuning process, model is coerced into adapting to misleading feature characteristics. This process causes irreversible deformation of the decision boundary, severely compromising model's ability to recall previous knowledge.

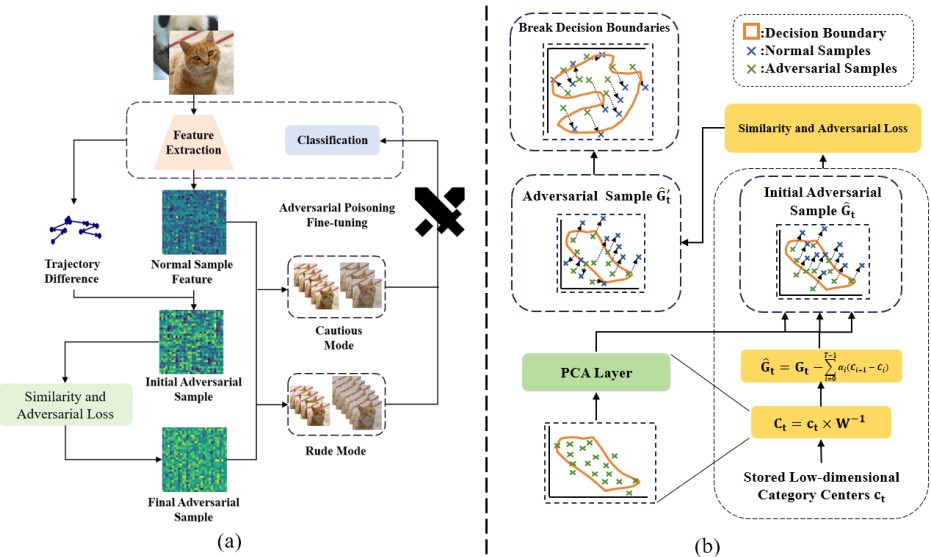

(a)                                     (b)

Figure 2: (a) Algorithm framework: adversarial examples are generated and optimized. Mixed datasets (with cautious/rude mixing ratios) are used for adversarial fine-tuning. (b) Detailed flow of adversarial example generation and decision boundary deformation.

Our fundamental insight for adversarial defense is that attacks exploit the instability of decision boundaries in continual learning. Therefore, an effective defense mechanism must adopt a multi-faceted strategy that proactively isolates threats, which is achieved by implementing a binary classifier with an elastic threshold. Concurrently, dynamic consistency constraints are applied to stabilize the decision boundary and cross-category comparison learning is employed to reinforce the feature space by strengthening intra-class compactness and inter-class separability.

### 3.2 ADVERSARIAL ATTACK WORKFLOW

The adversarial attack framework introduced in this paper consists of three integral components: (1) adversarial sample generation, (2) adaptive poisoning dataset strategy, and (3) adversarial fine-tuning. The overall architecture of the proposed method is depicted in Fig. 2(a).

**Step 1: Adversarial Sample Generation with Trajectory Weighting and Dual-Objective Optimization.** The first step focuses on crafting adversarial samples that are both effective and imperceptible. We propose a novel method based on trajectory difference weighted by a task-decaying factor, which accounts for the attenuation of past task influence over CL phases. Moreover, this study introduces the dual-objective optimization loss function which can retain similarity to the original input while reducing classification accuracy. The hyperparameter $\lambda$ controls the trade-off between attack strength and stealth. As a result, the majority of crafted samples lie outside the decision boundary of their original class, yet remain visually coherent with benign samples.

**Step 2: Adaptive Poisoning Dataset under Cautious and Rude Modes.** The generated adversarial samples are mixed with normal samples to form a composite dataset used for poisoning the fine-tuning process. The mixing ratio is critical for balancing attack potency and detectability. We introduce two distinct mixing modes in this study. The selection of mode can be made dynamically based on the attacker's objective and the defensive capabilities of target system. The mixing ratio used in cautious mode is 1:9. This mode prioritizes stealth, making the attack less likely to trigger anomaly detection mechanisms. It is suitable for scenarios requiring long-term persistence. The mixing ratio used in rude mode is 3:7, aiming for immediate and pronounced degradation of model

performance. It is effective in contexts where the primary goal is maximal disruption within a short time frame.

**Step 3: Adversarial Fine-Tuning for Inducing Irreversible Decision Boundary Deformation.** In the final step, the feature extraction layer is frozen to confine the impact of attack to the classifier layers. The mixed dataset $D_{mix}$ is then used to finetune the classifier. This process forces model to adapt to a combination of legitimate and malicious feature distributions, thereby distorting its decision boundaries in an irreversible manner. The loss function during fine-tuning is:

$$L_{ft} = \mathbb{E}_{(x,y) \sim D_{mix}} [l(h_\theta(x), y)] \tag{1}$$

As a consequence, the model not only misclassifies adversarial samples but also exhibits significantly degraded accuracy on benign samples from previous tasks. The entire process, summarized in Algorithm 1, operates effectively even with limited model access, highlighting its applicability in black-box and grey-box settings.

---

**Algorithm 1** Algorithm of Adversarial Attack

---

**Input:** Normal sample features $G_t$ in high-dimensional space, class centers $c_t$ obtained during training, trained classifier $F$, number of epochs $E$, task decay factor $\gamma$ ($\gamma \in (0, 1)$), trade-off parameter $\alpha$, initial model parameter $\theta$, a small constant preventing division by zero $\epsilon$.

**Output:** Adversarial sample $\hat{G}_t$ in high-dimensional feature space, adaptive poisoning dataset $\mathbb{D}_t$, model parameter $\theta_f$ after poisoning.

1: Load class centers $c_t$ and apply PCA to obtain $C_t$
2: Compute trajectory weights $\alpha_i \leftarrow \gamma^{N-1-i}$ for $i = 0$ to $N-1$
3: $\hat{G}_t \leftarrow G_t - \sum_{i=0}^{N-1} \alpha_i (C_{i+1} - C_i)$
4: **for** epoch = 1 to $E$ **do**
5:      $y' \leftarrow F(\hat{G}_t)$
6:      $L_{sim} \leftarrow \dfrac{\|\hat{G}_t - G_t\|_2}{\dim(G_t)}$, $L_{adv} \leftarrow L_{CE}(y', y_{wrong}) - L_{CE}(y', y_{true})$
7:      Compute gradients $\nabla_{\hat{G}_t} L_{sim}$, $\nabla_{\hat{G}_t} L_{adv}$
8:      $\nabla_{\hat{G}_t} L_{target} \leftarrow \nabla_{\hat{G}_t} L_{sim} + \nabla_{\hat{G}_t} L_{adv} + \epsilon$
9:      $\lambda_{sim} \leftarrow \dfrac{\|\nabla_{\hat{G}_t} L_{sim}\|}{\|\nabla_{\hat{G}_t} L_{target}\|}$, $\lambda_{adv} \leftarrow \dfrac{\|\nabla_{\hat{G}_t} L_{adv}\|}{\|\nabla_{\hat{G}_t} L_{target}\|}$
10:      $L \leftarrow \alpha \cdot \dfrac{L_{sim}}{\lambda_{sim}} + (1 - \alpha) \cdot \dfrac{L_{adv}}{\lambda_{adv}}$
11:      $\hat{G}_t \leftarrow \hat{G}_t - \eta \nabla_{\hat{G}_t} L$
12: **end for**
13: $\mathbb{D}_t \leftarrow$ mix up $G_t$ and $\hat{G}_t$
14: $\theta_f \leftarrow$ poison $\theta$ based on $\mathbb{D}_t$

---

### 3.3 TECHNICAL DETAILS

#### 3.3.1 ADVERSARIAL ATTACK

Fig. 2(b) illustrates the adversarial sample generation and the decision boundary deformation process proposed in this study. During model training, we retain only the dimensionality-reduced category centers $c$, which significantly reduces storage requirements while maintaining essential semantic information. In the adversarial sample generation stage, the model first acquires features of categories, then computes dimensionality reduction matrices using PCA to project the category centers $c$ back into the high-dimensional feature space to obtain the category center $C$.

The initial adversarial sample $\hat{G}_i$ is constructed by injecting a weighted trajectory difference of category centers into the high-dimensional feature $G_i$. The trajectory term, $\sum_{i=1}^{T-1} \alpha_i (C_{i+1} - C_i)$, captures the historical drift of category centers across sequential tasks, effectively demonstrating the inherent instability of CL process. The weight coefficients $\alpha_i$ follow an exponential decay schedule. The design reflects the attenuated influence of earlier tasks while maintaining attack stealthiness. The adversarial feature is thus initialized with the specific formula referenced in Algorithm 1.

While decision boundary drift reduces classification accuracy, using it alone is often insufficient to induce comprehensive misclassification. To enhance the effectiveness and reliability of attack, we introduce similarity loss and adversarial loss. The similarity loss constrains adversarial samples from deviating too far from original samples in feature space, thereby maintaining perceptual consistency and attack stealth. The adversarial loss aims to maximize the model's confidence in incorrect labels (we use the label with the second highest probability in the original prediction distribution specifically). The overall adversarial loss function combines two components through an adaptive weighting mechanism, with the specific formula referenced in Algorithm 1.

To intelligently balance these competing objectives during optimization, we introduce a gradient norm normalization strategy. The normalization coefficients $\lambda_{sim}$ and $\lambda_{adv}$ are computed in real-time based on the current gradient magnitudes, with the specific formula referenced in Algorithm 1. This adaptive normalization scheme effectively mitigates potential gradient conflicts and ensures stable optimization throughout the adversarial sample generation process. Unlike traditional adversarial loss functions, our approach enables dynamic and real-time gradient balancing, significantly improving the convergence properties and effectiveness of the generated samples

### 3.3.2 ADVERSARIAL DEFENSE

Next, we will provide a detailed introduction to the specific design of this study regarding resilient rejection, dynamic consistency constraint, and cross-category comparison learning. The specific algorithm is shown in Alg.2. The specific experimental results are presented in Appendix C.

---

**Algorithm 2** Algorithm of Adversarial Defense

**Input:** Normal sample $x$, gaussian noise $\epsilon$ of the same size as $G_t$, number of epochs $E$, number of samples $N$, initial model parameter $\theta$, learning rate $\eta$
**Output:** Defened model parameter $\theta$
1: Load initial model parameter $\theta$
2: **for** epoch = 1 to E **do**
3:     $\hat{x} = x + \epsilon$
4:     $L_{feat} = \|F(\hat{x}) - F(x)\|_2$
5:     Compute $L_{rej}$ and $L_{com}$ based on (2) and (3)
6:     $L(\theta) = \alpha L_{feat} + \beta L_{com} + \gamma L_{rej}$
7:     $\theta \leftarrow Adam(\theta, L(\theta), \eta)$
8: **end for**

---

As shown in Fig.2(a), we use normal samples with randomly added noise to train the binary classifier $C_d$ and output anomaly score. In addition, a dynamic threshold strategy is introduced to adaptively adjust the rejection boundary by combining the local feature consistency and global uncertainty of the samples. The specific formula is as follows:

$$L_{rej} = \frac{\alpha'}{N} \sum_{i=1}^{N} -log \frac{exp(F(x_i))}{\sum_{k=0}^{i} exp(F(x_k))} + \frac{1-\alpha'}{N} \sum_{i=1}^{N} (z_i^j(m+s_i) + (1-z_i^j)(m-s_i)) \quad (2)$$

where $z_i^j$ represents sample label ($z_i^j = 1$: Normal, $z_i^j = 0$: Rejection), $m$ represents rejection margin and rejection score $s_i$ for the $i$-th sample.

The core objective is to limit the deformation of decision boundary in the feature space. By applying differentiation constraints to model (especially the last layer of feature extraction module), the differentiation of adversarial samples from original samples is restricted, and $L_2$ distance between adversarial and normal samples is added to the loss function optimization model, which strengthens the model's ability to defend against adversarial attacks and improves the model's robustness.

Cross-category comparison learning is introduced to enhance feature aggregation within categories. Specifically, adversarial samples are treated as difficult positive samples while samples of other categories as negative samples. By designing loss function, we hope that feature distances between normal and adversarial samples are less than distance between different categories of samples, which can enhance intra-class compactness. In addition, it is also able to solve problem of representation forgetting that can occur in continual learning. The specific formula is as follows:

$$L_{com} = -\frac{1}{2N} \sum_{i=1}^{N} log \frac{\sum_{j \in P_i} exp(cos(F(x_i) - F(\hat{x}_j))/\tau)}{\sum_{k \in N_i} exp(cos(F(x_i) - F(x_k))/\tau)} \quad (3)$$

where $P_i$ is homogeneous set of sample $i$ (including adversarial samples), $N_i$ is heterogeneous set of sample $i$ and the temperature coefficient $\tau$.

## 4 EXPERIMENTAL RESULTS

In this section, we discuss the evaluation results of the above methods. The experiments are based on multiple basic image datasets such as CIFAR-100 Krizhevsky et al. (2009), using multiple evaluation metrics such as BWT and ACC, and our algorithm is compared with various continuous learning algorithms. The specific implementation settings are shown in Appendix A. We progressively verify the effectiveness of the adversarial attack algorithm through four experiments. Based on Table 1, we can find that only using the trajectory difference as the initial perturbation can significantly reduce the model performance. Then, we use multiple indicators to evaluate the quality of adversarial samples, the specific results can be found in Appendix B. Finally, Table 3 proves that the attack mechanism has universal destructive power on different CL algorithms and model architectures.

### 4.1 EFFECTIVENESS OF ADVERSARIAL SAMPLES

To validate the effectiveness of adversarial samples, Table 1 shows the classification accuracies of adversarial samples from different datasets on the Resnet-16 model trained using different CL algorithms. No subsequent optimization indicates the initial adversarial samples generated by increasing the trajectory difference only to the normal samples, while with(without) error labels indicates the samples generated by (not) fixing the error labels.

From Table 1, we can find that even the initial adversarial samples generated solely based on category-centered trajectory differences lead to significantly reduced classification accuracy compared to normal samples, with an average decrease of 9.79% across all datasets and CL algorithms. On the other hand, the effectiveness of adversarial attacks varies substantially across different CL algorithms. For instance, on CIFAR-100, MAS and RWALK show higher vulnerability to initial attacks (79.94% and 82.94% ASR) compared to EWC and AFEC (56.53% and 55.9% ASR).

The adversarial samples generated under the "with error labels" condition demonstrate significantly stronger attack effectiveness, reducing classification accuracy by an average of 27.01% compared to normal samples. Moreover, under the "without error labels" condition, the classification accuracy for adversarial samples reaches the lowest among all three scenarios with an average decrease of 30.19%, and ASR values approach near-perfect attack success rates (97.67%-99.93% across different configurations), indicating that these adversarial samples are almost indistinguishable from random noise to the target model. These results demonstrate that our adversarial sample generation method effectively compromises the robustness of continual learning models.

Table 1: Effectiveness of Adversarial Samples

| Datasets | CL Algorithms | ACC (%) | No subsequent optimization | | With error labels | | Without error labels | |
|---|---|---|---|---|---|---|---|---|
| | | | ASR (%) | TCC | ASR | TCC | ASR | TCC |
| CIFAR-100 | EWC | 60.2 | 56.53 | 0.55 | 95.4 | 0.72 | 99.43 | 0.78 |
| | AFEC | 64.7 | 55.9 | 0.54 | 90.21 | 0.68 | 98.87 | 0.75 |
| | MAS | 24.4 | 79.94 | 0.62 | 97.9 | 0.74 | 99.69 | 0.79 |
| | RWALK | 17.9 | 82.94 | 0.64 | 96.59 | 0.73 | 99.49 | 0.78 |
| Mini-ImageNet | EWC | 44.7 | 78.83 | 0.61 | 97.9 | 0.74 | 99.67 | 0.79 |
| | AFEC | 41.1 | 77.86 | 0.60 | 90.13 | 0.68 | 99.09 | 0.76 |
| | MAS | 20.6 | 88.41 | 0.67 | 98.73 | 0.76 | 99.93 | 0.81 |
| | RWALK | 15.2 | 84.93 | 0.65 | 98.6 | 0.75 | 99.87 | 0.80 |
| Tiny-ImageNet | EWC | 33.12 | 77.44 | 0.60 | 98.44 | 0.75 | 99.68 | 0.79 |
| | AFEC | 30.4 | 79.29 | 0.62 | 95.08 | 0.71 | 97.67 | 0.74 |
| | MAS | 9.8 | 91.57 | 0.69 | 98.48 | 0.75 | 99.61 | 0.79 |
| | RWALK | 7.58 | 94.19 | 0.71 | 96.89 | 0.72 | 99.55 | 0.78 |

### 4.2 EVALUATION OF ADVERSARIAL ATTACKS

This study evaluates adversarial attack effectiveness by fine-tuning trained models with mixtures of adversarial and normal samples, employing different mixing ratios to distinguish attack modes. The cautious mode uses a 1:9 ratio of adversarial to normal samples, while the rude mode employs a more aggressive 3:7 ratio. For comparison, we include a noise mode where 30% of normal samples

are randomly selected and noise is added. The detailed results presented in Table 2 demonstrate both proposed adversarial attack approaches significantly outperform the random noise approach across all datasets. The rude mode reduces accuracy to 15.86% on CIFAR-100 compared to 47.15% for the noise mode, representing a 66.4% improvement in attack effectiveness. The rude mode consistently produces the strongest attacks, reducing accuracy by an average of 75.5% across datasets compared to normal samples, while also causing the most substantial negative BWT values. Moreover, dataset complexity influences attack effectiveness, with more complex datasets (Tiny-ImageNet) demonstrating greater vulnerability to adversarial attacks.

Table 2: Experimental Results of Adversarial Attack

| Dataset | Normal Samples | | Cautious Mode | | Rude Mode | | Noise Mode | |
|---|---|---|---|---|---|---|---|---|
| | BWT (%) | ACC (%) | BWT (%) | ACC (%) | BWT (%) | ACC (%) | BWT (%) | ACC (%) |
| CIFAR-100 | -2.90 | 64.70 | -31.62 | 25.06 | -40.38 | 15.86 | -8.70 | 47.15 |
| Mini-ImageNet | -1.30 | 41.10 | -21.43 | 27.78 | -37.40 | 10.10 | -7.31 | 40.51 |
| Tiny-ImageNet | -1.30 | 30.40 | -23.22 | 5.11 | -23.34 | 4.41 | -21.11 | 8.60 |

## 4.3 EVALUATION OF ATTACK EFFECTIVENESS FOR OTHER MODELS

This study presents a comprehensive evaluation of the proposed adversarial attack methodology across five diverse model architectures: VGGNet, Vision Transformer (ViT), ConvNext, Efficient-Next and InfLoRA. The integrated results in Table 3 reveal that while InfLoRA demonstrates exceptional performance on clean samples (97.42%), it exhibits remarkable susceptibility to sophisticated attacks, particularly without error labels (99.83% ASR). In addition, the progressive effectiveness of attack strategies follows a consistent pattern across architectures. "Without error labels" consistently outperforms other strategies, achieving ASR values above 97% for most architectures. The significant performance gap between adversarial attacks and random noise underscores the targeted nature of the proposed methodology. While ViT-based architectures (ViT and InfLoRA) show relatively higher resistance to intermediate attacks, they remain vulnerable to sophisticated strategies.

Table 3: Comprehensive Evaluation of Adversarial Attack Across Model Architectures

| Model | ACC | ASR | | | ACC | | |
|---|---|---|---|---|---|---|---|
| | | A | B | C | Cautious Mode | Rude Mode | Noise Mode |
| VGGNet | 55.15 | 52.12 | 84.15 | 97.34 | 31.28 | 17.22 | 42.93 |
| ViT | 51.01 | 50.14 | 80.17 | 66.86 | 45.11 | 33.14 | 48.10 |
| ConvNext | 31.13 | 74.75 | 91.31 | 97.93 | 17.96 | 8.32 | 13.37 |
| EfficientNext | 37.11 | 77.14 | 85.69 | 99.07 | 28.95 | 14.80 | 32.74 |
| InfLoRA | 97.42 | 19.87 | 68.58 | 99.83 | 62.18 | 41.64 | 50.15 |

A:No subsequent optimization; B:With error labels; C:Without error labels

The consistent success of "without error labels" across all architectures suggests that it exploits fundamental weaknesses in CL rather than architecture-specific flaws. The minimal performance gap between cautious and rude mode for some architectures suggests that attack detectability may not necessarily increase with attack strength. The near-perfect success rates indicate that current approaches to adversarial defense remain inadequate against sophisticated attack methodologies.

## 5 RELATED WORK

**Continual Learning:** Continual learning approaches can be broadly classified into three main categories: regularization-based algorithms (Kirkpatrick et al. (2017); Li & Hoiem (2017); Mai et al. (2022); Wu et al. (2019)), rehearsal-based approachesLi & Hoiem (2017); Aljundi et al. (2019); Chaudhry et al. (2018); Gao & Liu (2023); Jiang et al. (2021); Hou et al. (2019); Dong et al. (2023; 2021); Wang et al. (2022b;c); Buzzega et al. (2020); Cha et al. (2021); Shokri & Shmatikov (2015), architectural-based approachesYoon et al. (2017); Fernando et al. (2017); Serra et al. (2018); Yan et al. (2021); Li et al. (2021). In recent years, CL researches (Douillard et al. (2022); Wang et al. (2022a); Gao et al. (2023); Chen et al. (2025); Liu & Chang (2025a); Zheng et al. (2025)) have fo-

cused on achieving efficient resource utilization while mitigating catastrophic forgetting. A detailed introduction to CL algorithms is presented in Appendix D.

**Adversarial Attack and Defense:** Adversarial attack includes two forms: white box attack and black box attack. White-box attack (Goodfellow et al. (2014); Kurakin et al. (2016); Madry et al. (2017); Carlini & Wagner (2017); Moosavi-Dezfooli et al. (2016)) assumes that attacker is fully informed about the structure and parameters of model. Black-box attack (Chen et al. (2017); Carlini & Wagner (2017); Ilyas et al. (2018); Su et al. (2019); Papernot et al. (2017)) assumes that attacker has no direct access to the structure or parameters of target model. Adversarial defense includes various methods such as adversarial training (Goodfellow et al. (2014); Madry et al. (2017); Kurakin et al. (2018); Athalye et al. (2018); Ding et al. (2018)), randomization (Xie et al. (2017); Liu et al. (2018a;b)), generative models (Meng & Chen (2017); Samangouei et al. (2018); Li et al. (2018)) and adversarial detection (Gong & Wang (2023); Metzen et al. (2017); Zheng & Hong (2018); Roth et al. (2019); Yang et al. (2020); Feinman et al. (2017)). A detailed introduction to adversarial attacks and defense algorithms is presented in Appendix D.

**Adversarial Attacks in CL Scenarios:** The dynamic nature in CL system makes it particularly sensitive to adversarial sample attacks and can significantly lead to catastrophic forgetting by mixing adversarial samples and the training samples. BrainWashAbbasi et al. (2024) proposed a two-layer optimized poisoning strategy to force model to forget knowledge of previous tasks by contaminating the current task data. Li et al. Li & Ditzler (2022) proposed a task-level targeted poisoning attack to force model to forget knowledge of previous tasks by designing covertly clean-labeled samples in both domain-incremental and task-incremental learning scenarios. PACOLLi & Ditzler (2023) destroys model's ability to generative rehearsal and regularization methods by covert samples with limited perturbations. Kang et al.Kang et al. (2023) proposed an attack method against generative rehearsal that accelerates model forgetting by contaminating the generative playback samples, which significantly weakens the model performance at the dual level of generator and classifier.

# 6 CONCLUSION

This paper investigates the adversarial robustness of CL models in dynamic environments, systematically exploring adversarial attacks and defense mechanisms. Based on the observed decision boundary drift phenomenon, we design feature-level adversarial attack algorithms. By introducing concepts such as similarity loss, resilient rejection mechanisms and cross-category comparison learning, we establish an adversarial attack robustness framework and validate the effectiveness of framework.

However, this study has limitations, such as high computational costs and limited practical application scope. In the future, we will introduce lightweight rejection modules with shared parameters and knowledge distillation techniques to reduce model size and computational overhead. In addition, we will incorporate reinforcement learning to enhance the resilient rejection mechanism, enabling dynamic adjustment of defense strategies. Additionally, future research will explore adversarial attack and defense methods for CL models in multi-modal scenarios and investigate the transferability of defense capabilities across industry sectors.

# 7 ETHICAL CONSIDERATIONS

Our research aims to proactively expose and understand the vulnerabilities of CL systems to adversarial attacks, with the ultimate goal of fostering the development of more robust and trustworthy AI systems. Similar to studies in security and adversarial machine learning, the evasion techniques we present could potentially be misused to compromise CL systems. We firmly believe that responsible disclosure of these vulnerabilities, coupled with the development of effective defenses, is paramount for the long-term health and security of the AI ecosystem. Our research utilizes only publicly available benchmark datasets (CIFAR-100, Mini-ImageNet, Tiny-ImageNet) that contain non-personal, non-identifying image data. Therefore, our work does not raise concerns regarding privacy or data protection.

## THE USE OF LARGE LANGUAGE MODELS (LLMS)

In preparing this manuscript, we used LLMs to assist in language polishing and improving readability of the text. The LLMs were used solely for drafting and refining phrasing; all scientific content, ideas, derivations, and experimental results were developed and verified by the authors. We take full responsibility for the accuracy, validity, and integrity of all content in this manuscript, including any portions generated with the assistance of LLMs.

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

## A  EXPERIMENT SETUP

### A.1  DATASETS

In our experiments, we evaluate the algorithm performance using several datasets: the CIFAR-100(Krizhevsky et al. (2009)), Mini-ImageNet(Vinyals et al. (2016)), and Tiny-ImageNet(Le & Yang (2015)):

- CIFAR-100: It consists of 100 image categories and contains 60,000 32×32 pixel color images. Compared to other datasets, CIFAR-100 has a small image size, which makes it more suitable as a benchmark test dataset for lightweight models.

- Mini-ImageNet: It is a 100 categories of images subset extracted from the ImageNet dataset, containing a total of 60,000 84×84 pixel color images. Compared with the original dataset, the extracted subset is more suitable for small sample scenarios and requires less computational resources.

- Tiny-ImageNet: it is a dataset containing 200 categories of images and contains a total of 100000 64×64 pixel color images. Compared to the above datasets, this dataset has more samples and complex categories.

### A.2  MODEL SETTING

We use ResNet-18 (He et al. (2016)) as the base model for the study (unless otherwise stated) and choose stochastic gradient descent method to optimize model which uses a learning rate of 0.01 and its batch size is set to 16. All images are first normalized in the range of [0,1], and the antagonistic sample data is truncated to this range. In the experiments of this study, we tested classical CL algorithms such as Elastic weight consolidation (EWC,Kirkpatrick et al. (2017)), Memory aware synapses (MAS,Aljundi et al. (2018)) and novel CL algorithms such as Active forgetting of negative transfer (AFEC,Wang et al. (2021)).

### A.3  METRICS

**Backward Transfer (BWT):** This work employs BWT (Lopez-Paz & Ranzato (2017)) to quantify the model's predictive performance on previous tasks after learning new ones. BWT serves as an indicator of catastrophic forgetting: a lower BWT value corresponds to greater forgetting of earlier tasks. Accordingly, the adversarial objective in this study is to minimize the BWT score, thereby intentionally exacerbating forgetting in the model. Let $A_{t,i}$ denote the classification accuracy of the

continual learning model on task $i$ after training on task $t$, where $t > i$. The BWT score is formally defined as:

$$BWT = \frac{1}{T-1}(A_{T,i} - A_{i,i}) \tag{4}$$

**Classification Accuracy (ACC):** This work employs ACC as a fundamental metric to quantify the overall predictive performance of the model on a given task. ACC measures the proportion of correctly classified samples among all evaluated samples. A higher ACC value indicates better overall performance of the model. Formally, let $N_{\text{correct}}$ denote the number of correctly classified samples and $N_{\text{total}}$ denote the total number of samples in the evaluation set. The ACC is formally defined as:

$$ACC = \frac{N_{\text{correct}}}{N_{\text{total}}} \times 100\% \tag{5}$$

**Attack Success Rate (ASR):** This work employs ASR to quantify the effectiveness of adversarial attacks. ASR measures the proportion of adversarial examples that successfully cause the model to misclassify. A higher ASR value indicates a more potent attack. Formally, let $N_{\text{total}}$ denote the total number of original samples correctly classified, and $N_{\text{success}}$ denote the number of corresponding adversarial examples that are misclassified. The ASR is defined as:

$$ASR = \frac{N_{\text{success}}}{N_{\text{total}}} \times 100\% \tag{6}$$

**Target Class Confidence (TCC):** This work employs TCC as a metric to quantify the model's certainty in its erroneous predictions caused by adversarial examples. A higher TCC value indicates a more confident and potentially more robust attack. For a targeted attack where $y_{\text{target}}$ is the desired wrong label, and for a non-targeted attack where $y_{\text{wrong}}$ is the model's predicted class (different from the true label), the TCC is formally defined as:

$$TCC = F(x')[y_{\text{adv}}] \tag{7}$$

where $F(x')$ represents the model's output probability distribution for the adversarial example $x'$, and $y_{\text{adv}}$ denotes either $y_{\text{target}}$ or $y_{\text{wrong}}$ depending on the attack type.

**Mean Square Error (MSE)**: This work employs MSE as a metric to calculate the pixel-level difference between the restored image and the normal sample. Lower values indicate higher similarity and better adversarial sample quality. The MSE is defined as:

$$MSE = \frac{1}{n}\sum_{i=1}^{n}(x_i - y_i)^2 \tag{8}$$

where $x_i$ and $y_i$ represent pixel values of the restored and normal images respectively.

**Cosine Similarity**: This work employs cosine similarity as a metric to measure the directional similarity between image feature vectors in high-dimensional space. Values closer to 1 indicate higher similarity. The cosine similarity is defined as:

$$CosineSimilarity = \frac{\mathbf{A} \cdot \mathbf{B}}{\|\mathbf{A}\|\|\mathbf{B}\|} \tag{9}$$

where $\mathbf{A}$ and $\mathbf{B}$ are feature vectors of the compared images.

Structural Similarity (SSIM): This work employs SSIM to comprehensively evaluates image similarity from three aspects: luminance, contrast, and structure. Values closer to 1 indicate higher perceptual similarity. The SSIM index is defined as:

$$SSIM(x,y) = \frac{(2\mu_x\mu_y + c_1)(2\sigma_{xy} + c_2)}{(\mu_x^2 + \mu_y^2 + c_1)(\sigma_x^2 + \sigma_y^2 + c_2)} \tag{10}$$

where $\mu$ represents mean intensity, $\sigma$ represents standard deviation, and $c_1$, $c_2$ are stabilization constants.

Perceptual Feature Distance(PFD): This work employs perceptual feature distance(PFD) to extract deep features using a pre-trained VGG network (Simonyan & Zisserman (2014)) and computes the $L_2$ distance between feature representations of compared images. Lower distances indicate higher semantic-level similarity. The perceptual distance is calculated as:

$$\text{PFD} = \|\phi(x) - \phi(y)\|_2 \tag{11}$$

where $\phi(\cdot)$ represents the feature extraction function of the VGG network.

## B    Quality Assessment of Adversarial Samples

Since adversarial samples in this study are essentially high-dimensional features corresponding to images and lack direct visualization information, it is difficult to directly judge whether the generated adversarial samples meet the basic definition of "indistinguishable by human eyes". Therefore, this study introduces a diffusion model as an auxiliary means to determine whether the adversarial samples satisfy this basic requirement. We fine-tune the diffusion model using normal training samples and their corresponding high-dimensional features, then feed the adversarial samples into this model to obtain restored images for visual assessment. To quantitatively evaluate the visual consistency between restored images and normal samples, this study employs multiple metrics in A with distinct characteristics.

Table 4: Validation of Quality Assessment of Adversarial Samples

| Dataset | Adversarial Sample | MSE | Cosine Similarity | SSIM | PFD |
|---------|--------------------|----|-------------------|------|-----|
| CIFAR-100 | No subsequent optimization | 69.82 | 0.812 | 0.88 | 277.15 |
| | With error labels | 80.98 | 0.685 | 0.61 | 380.03 |
| | Without error labels | 212.2 | 0.347 | 0.43 | 783.15 |
| Mini-ImageNet | No subsequent optimization | 86.11 | 0.704 | 0.79 | 398.33 |
| | With error labels | 107.65 | 0.498 | 0.55 | 521.69 |
| | Without error labels | 362.37 | 0.233 | 0.31 | 971.20 |
| Tiny-ImageNet | No subsequent optimization | 87.11 | 0.761 | 0.82 | 317.51 |
| | With error labels | 130.82 | 0.553 | 0.70 | 530.63 |
| | Without error labels | 265.94 | 0.309 | 0.37 | 780.24 |

The comprehensive results presented in Table 4 reveal that adversarial samples generated with only category-centered trajectory differences ("No subsequent optimization") show minimal deviation from normal samples across all metrics, with average MSE of 81.01, cosine similarity of 0.759, structural similarity of 0.83, and perceptual distance of 330.99 across datasets. After training with error labels, all metrics indicate increased deviation from normal samples, with average increases of 33.2% in MSE, 27.8% reduction in cosine similarity, 26.5% reduction in structural similarity, and 47.5% increase in perceptual distance compared to the initial adversarial samples. Moreover, adversarial samples generated without error labels demonstrate the most significant deviations, with metrics approaching values similar to random noise. These values represent increases of 123.7% in MSE, 61.0% reduction in cosine similarity, 55.4% reduction in structural similarity, and 155.3% increase in perceptual distance compared to the initial adversarial samples.

These quantitative findings confirm that while the initial adversarial samples maintain high visual similarity to original images (making them difficult to distinguish by human eyes), the training process significantly increases their divergence. The structural similarity metric, being particularly aligned with human visual perception, provides strong evidence that the "with error labels" samples remain visually plausible while achieving effective adversarial properties.

## C    Adversarial Defense Experiment

### C.1    Experimental results

To better test the adversarial defense method in this study, we add random noise to the normal samples to generate adversarial samples to fine-tune the initialization model. The following figure 3 shows the changes in the high-dimensional features of the image before and after fine-tuning the model. From the changes in the sample features before and after fine-tuning, we can find that the feature extraction ability of the model rises significantly after using the algorithm in this study. It is worth noting that we believe that the confusion of features in Fig.3 is not entirely due to the weak classification ability of the model, but more due to the existence of intersecting decision boundaries, resulting in the situation of confusion of the low-dimensional features; while the order in Fig.3(b) is also due to the strong classification ability of the model, the decision boundaries are clearly defined, and the same category of images are better able to be clustering.

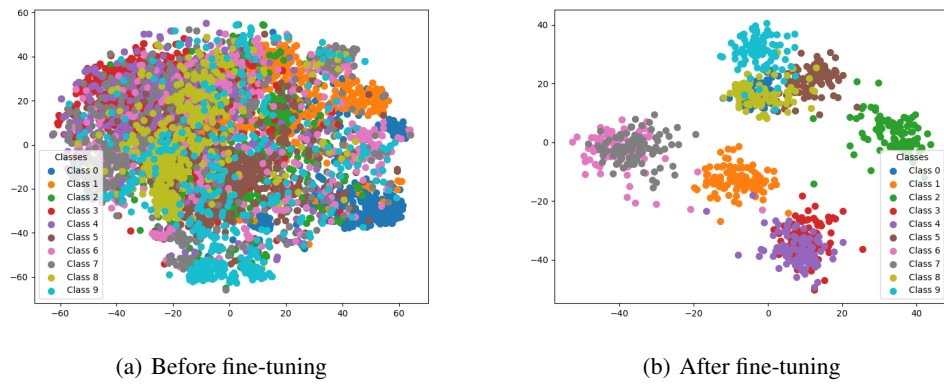

(a) Before fine-tuning         (b) After fine-tuning

Figure 3: Visualization of dimensionality reduction features before and after fine-tuning

Table 5: Classification Accuracy at different stages of model

|  | Classification Accuracy | Rejection Rate |
|---|---|---|
| Normal sample before fine-tuning | 64.7 | — |
| Fine-tuned normal sample | 79.58 | 3.52 |
| Adversarial sample | 66.25 | 40.16 |

The classification accuracy calculation in Table 5 excludes the rejected samples first, so the classification accuracy will be relatively higher. In addition, this study found that the model classification performance improves after fine-tuning, which is due to the fact that the decision boundary is clearer and more explicit after fine-tuning, which will also lead to the probability of the previously generated adversarial samples even if they do not fall within the decision boundary, than the probability of misclassification.

## C.2 Ablation Experiments

From Table 6, it can be seen that when dynamic consistency constraint is removed, the accuracy of both normal and adversarial samples decreases, but the decrease for adversarial samples is more significant. This is because after removing dynamic consistency constraint mechanism, the difference between adversarial samples and normal samples expands, and adversarial samples are more likely to successfully deceive model. In addition, the rejection rate of normal samples fluctuates greatly compared with adversarial samples, which is due to the stability of decision boundaries and the consistency of normal sample features decreases. When cross-category comparison learning is removed, the intra-class compactness is weakened and the classification accuracy of normal and adversarial samples decreases. When the dynamic rejection mechanism is removed, the accuracy of normal samples increases slightly. This is due to the fact that when the rejection mechanism is removed, all samples are forced to be classified. In addition, due to the lack of protection from the rejection mechanism, the accuracy of model against adversarial samples is significantly reduced.

## D Related Work

### D.1 Continual Learning

Continual learning enables models to cope with non-stationary data in dynamic environments by using an updating mechanism that maintains historical task performance while absorbing new knowledge. CL approaches can be broadly classified into three main categories: Regularization-based algorithms (Kirkpatrick et al. (2017); Li & Hoiem (2017); Mai et al. (2022); Wu et al. (2019)) penalize changes in network parameters of the current task with respect to the previous task, which

Table 6: The impact on performance after eliminating modules

| A | B | C | Normal Sample | | Adversarial Sample | |
|---|---|---|---|---|---|---|
| | | | Accuracy(%) | Rejection(%) | Accuracy(%) | Rejection(%) |
| ✗ | ✓ | ✓ | 65.53 | 11.02 | 21.05 | 39.68 |
| ✓ | ✗ | ✓ | 63.20 | 8.75 | 41.59 | 45.97 |
| ✓ | ✓ | ✗ | 83.72 | — | 19.27 | — |
| ✓ | ✓ | ✓ | 79.58 | 3.52 | 66.25 | 40.16 |

A:Dynamic Consistency Constraint Mechanism
B:Cross-Category Comparison Learning
C:Resilient Rejection Mechanism

limits model plasticity and thus solves the catastrophic forgetting problem. Rehearsal-based approaches (Li & Hoiem (2017); Aljundi et al. (2019); Chaudhry et al. (2018); Gao & Liu (2023); Jiang et al. (2021); Hou et al. (2019); Dong et al. (2023; 2021); Wang et al. (2022b;c); Buzzega et al. (2020); Cha et al. (2021); Shokri & Shmatikov (2015)) retain previous data by constructing a buffer of examples, which is then used to train the model alongside the new training dataset. Architectural-based approaches (Yoon et al. (2017); Fernando et al. (2017); Serra et al. (2018); Yan et al. (2021); Li et al. (2021)) construct task-specific parameters and allow network expansion during CL. The adversarial defense approach of this work focuses on regularization-based methods that optimize the model by designing a specific regularization loss function. Moreover, our adversarial attack method performs well in the above scenarios.

In recent years, CL researches have focused on achieving efficient resource utilization while mitigating catastrophic forgetting. In terms of model structure dynamization, DyTox (Douillard et al. (2022)) provides dedicated processing paths for each new task by dynamically expanding the task-specific token while maintaining the shared backbone. Meanwhile, S-Prompts Wang et al. (2022a)) introduces prompt-tuning into CL, combining it with the Transformer structure to achieve low-forgotten domain incremental learning. Additionally, LAE framework (Gao et al. (2023)) trains a fine-tuning parameter module for new tasks online, and integrates new knowledge into the offline master module. Various strategies have been proposed to alleviate forgetting and avoid storing raw data: STAR (Chen et al. (2025)) optimizes the loss function design to effectively solve the categories imbalance problem in zero-sample rehearsal, while LoRA Subtraction (Liu & Chang (2025a)) constructs a drift-resistant space and uses old category prototypes to assist new category learning, reducing the dependency on old tasks. In terms of feature representation and integration, TagFex (Zheng et al. (2025)) constructs a task-independent self-supervised model and introduces a merged-attention module to extend features which effectively prevents mis-classification during class incremental learning.

## D.2 ADVERSARIAL ATTACK

### D.2.1 WHITE-BOX ATTACK

White-box attack assumes that attacker is fully informed about the structure and parameters of model, and attacker generates adversarial samples based on the gradient of the loss function with respect to the inputs. FGSM (Goodfellow et al. (2014)) generates adversarial perturbations using the sign of the gradient with respect to inputs. Based on this, BIM (Kurakin et al. (2016)) generates more efficient adversarial samples by updating the gradient in multiple small steps. As a widely adopted baseline method, PGD (Madry et al. (2017)) uses stochastic initialization and projection operations to generate samples. In addition to gradient-based methods, the C&W attack (Carlini & Wagner (2017)) controls the size of perturbation and attack success by optimizing the objective function, while DeepFool (Moosavi-Dezfooli et al. (2016)) iteratively computes the minimum perturbation that moves the input out of the decision boundary.

### D.2.2 BLACK-BOX ATTACK

Black-box attack assumes that attacker has no direct access to the structure or parameters of target model, and it is now common to generate adversarial samples indirectly by using an agent model or observing the behavior of target model. ZOO (Chen et al. (2017)) uses finite difference method to estimate the gradient and applies the C&W (Carlini & Wagner (2017)) attack to generate the adversarial samples. Additionally, Natural Evolutionary Strategy (Ilyas et al. (2018)) estimates the gradient through a sampling strategy for more efficient generation. Single pixel attack (Su et al. (2019)) realizes the attack by modifying one pixel of the image. In addition, transfer attack (Papernot et al. (2017)) generates adversarial samples by observing the labels predicted by the model and using an agent model before attacking the target model.

### D.2.3 ADVERSARIAL DEFENSE

Adversarial defense include various methods such as adversarial training, randomization, generative models and adversarial detection. By generating adversarial samples (Goodfellow et al. (2014); Madry et al. (2017); Kurakin et al. (2018); Athalye et al. (2018); Ding et al. (2018)) and using examples to train model, the robust performance of the classifier against adversarial samples can be significantly improved. In addition, robustness can be improved by incorporating random components into the model to simulate adversarial perturbations (Xie et al. (2017); Liu et al. (2018a;b)). Defense mechanisms based on generative model (Meng & Chen (2017); Samangouei et al. (2018); Li et al. (2018)) can reduce the impact of adversarial perturbations on the model by projecting adversarial samples onto the manifold learned by generative model. Unlike the above approaches, adversarial detection (Gong & Wang (2023); Metzen et al. (2017); Zheng & Hong (2018); Roth et al. (2019); Yang et al. (2020); Feinman et al. (2017)) aims at recognizing the presence of adversarial attacks, with the core assumption that adversarial samples are distributed differently from the natural data and usually far away from the natural data manifold.

