# OpenReview forum: "An Adversarial Attack Framework via Decision Boundary Drift for Continual Learning"
_ICLR.cc/2026/Conference — ICLR 2026 Conference Withdrawn Submission_

### Official Review · Reviewer_XXnZ · 2025-10-31

**Soundness:** 2
**Presentation:** 2
**Contribution:** 2
**Rating:** 2
**Confidence:** 4

**Summary:**

This work proposes an adversarial attack framework, AADB, to address the risk posed by the weight drift phenomenon. The framework includes adversarial sample generation and defense mechanisms, such as feeding rejection loss and a cross-category comparison loss. The effectiveness of the adversarial samples in attacking continual learning (CL) is also validated.

**Strengths:**

1. Focused study on the effectiveness of adversarial attacks and their impact.

2. Detailed and clear presentation of the main body and results.

3. Systematic evaluation focused on the effectiveness of adversarial attacks.

**Weaknesses:**

### 1. Unelaborated Problem Set-Up
Weight drift, and the resulting feature drifts, are well-known and widely discussed phenomena in continual learning. This paper aims to address this problem and proposes a method, but it is unclear what the method solves distinctively compared to other approaches that handle drifts (e.g., memory, replay, regularization, external models, distillation, or even adversarial attacks on CL). As a result, it is also unclear how the method addresses potential limitations. If the goal is to narrow down the problem, the paper should justify why and what distinguishes its coverage from existing approaches.

### 2. Missing Analysis and Consideration
The evaluation relies solely on quantitative analysis of attack quality, which is insufficient to demonstrate weight drift mitigation. A more direct and deeper qualitative analysis is needed. Furthermore, decision boundary (DB) changes caused by adversarial attacks exhibit high randomness, leading to inconsistency of the derived DB changes on a high-dimensional space. Without clearly establishing the relationship between DB and adversarial samples, it is even more difficult to prove that weight drift is mitigated solely by the quality of the adversarial samples.

### 3. Issues in Validation
a. Similar to the point above, AADB is proposed to address weight drift, but the validation mostly focuses on the quality of adversarial samples rather than demonstrating resolution of weight drift. This creates a mismatch between the validation and the target problem.

b. This paper is more practical approach, so validation of its impact in real-world scenarios is necessary. Many continual learning (CL) benchmarks exist, but not evaluated in this paper.

c. The rationale for selecting only MAS, EWC, RWALK, and AFEC is not explained. These are regularizers rather than full CL algorithms. To properly compare drift effect resolution, additional approaches handling drift effect should be included. The current setup may give the impression that AADB is primarily a regularization method; if so, this should be explicitly stated rather than claiming a general weight drift resolution, which can be addressed from many perspectives.

d. The datasets used are not clearly described, and no links are provided (Section 4). Only CIFAR-100 is mentioned; if that is the only dataset, wider validation across additional datasets is required.

**Questions:**

.

---

> ### Author Response · Authors · 2025-11-21
>
> Thank you for your suggestions. I will carefully consider them and make improvements.

---

### Official Review · Reviewer_jHEW · 2025-10-31

**Soundness:** 2
**Presentation:** 2
**Contribution:** 2
**Rating:** 2
**Confidence:** 3

**Summary:**

The paper investigates adversarial robustness in CL and the core idea is to exploit the phenomenon of *decision boundary drift*—the shift of category centers during sequential learning—to construct feature-level adversarial samples. The framework consists of three stages: trajectory-weighted adversarial sample generation, adaptive poisoning via mixed datasets, and adversarial fine-tuning to distort the decision boundary. A complementary defense module is also proposed, combining dynamic consistency constraint, cross-category comparison learning, and resilient rejection.

**Strengths:**

- The topic is timely and relevant, addressing an emerging security concern in lifelong learning systems.
- The paper attempts to integrate both attack and defense perspectives within a unified conceptual framework, which could provide a useful foundation if developed more rigorously.

**Weaknesses:**

1. The paper’s structure is difficult to follow, and several sections (methodology and defense) lack clear mathematical definitions or logical flow, making the framework hard to interpret.
2. The proposed AADB largely extends existing poisoning or feature-drift attacks using heuristic weighting and lacks comparison with recent baselines.
3. Reported results show large accuracy drops but omit statistical analysis, visualization of adversarial samples, or ablations proving each component’s contribution.

**Questions:**

1. How is “decision boundary drift” formally measured or quantified beyond the qualitative plots in Figure 1?
2. In what essential way does AADB differ from prior continual learning poisoning attacks?
3. Could the authors analyze computational efficiency (e.g., training time, additional memory, and runtime overhead) of both attack and defense modules?

---

> ### Author Response · Authors · 2025-11-21
>
> Thank you for your suggestions. I will carefully consider them and make improvements.

---

### Official Review · Reviewer_XFv4 · 2025-10-31

**Soundness:** 2
**Presentation:** 2
**Contribution:** 2
**Rating:** 2
**Confidence:** 3

**Summary:**

The paper presents AADB. It leverages Decision Boundary Drift (DBD) to generate feature-space adversarial samples that gradually distort model boundaries. The framework introduces three stages: trajectory-based perturbation, mixed data poisoning under different aggressiveness levels, and fine-tuning to amplify drift.  Experiments on three datasets show attack effectiveness against several CL baselines.

**Strengths:**

1. The paper is competently executed.
2. The work draws attention to a meaningful issue: the vulnerability of CL models to decision drift attacks.
3. Experimental results cover several benchmark datasets and CL baselines.
4. The paper provides both attack and defense perspectives, showing awareness of broader implications.

**Weaknesses:**

1. DBD is qualitatively motivated but lacks a quantitative metric or visualization beyond feature-center shifts.
2. Key equations (e.g., dual-objective loss) are under-explained, reducing clarity. The manuscript is verbose and occasionally inconsistent in notation (e.g., Algorithm 2).
3. The manuscript repeats high-level ideas across sections and lacks concise mathematical formulation, which affects readability and technical clarity.
4. Improvements are small and without statistical validation.

**Questions:**

1. How exactly is decision boundary drift measured? Can you visualize or quantify it using class-center displacement or boundary distance metrics?
2. What is the individual effect of the trajectory weighting factor and the cautious/rude mixing strategy? Please provide ablation results.
3. The proposed loss components (similarity loss, adversarial loss, consistency loss, etc.) are not sufficiently ablated or justified. Could you provide ablation results on each loss term to demonstrate necessity?

---

> ### Author Response · Authors · 2025-11-21
>
> Thank you for your suggestions. I will carefully consider them and make improvements.

---

### Official Review · Reviewer_pEDz · 2025-11-03

**Soundness:** 3
**Presentation:** 3
**Contribution:** 2
**Rating:** 4
**Confidence:** 4

**Summary:**

This paper proposes an adversarial attack framework (AADB) based on the decision boundary drift phenomenon for continual learning (CL) models. The AADB framework consists of three core components: adversarial example generation with trajectory weighting and a dual-objective loss, an adaptive poisoning dataset strategy with different mixing ratios, and adversarial fine-tuning to induce irreversible decision boundary deformation. Experiments on multiple datasets and CL algorithms show that the attack significantly degrades model performance, while the defense can effectively intercept 40.16% of adversarial examples.

**Strengths:**

+ The attack pipeline is well-designed, particularly the introduction of the task-decaying factor and the adaptive dual-objective loss.
+ The method's effectiveness is validated across multiple datasets, CL algorithms (EWC, AFEC, MAS, etc.), and model architectures (VGG, ViT, ConvNext, etc.). The evaluation of adversarial example quality using multiple metrics (e.g., SSIM, PFD) is also detailed.
+ The paper is well-structured and logically clear.

**Weaknesses:**

- The experimental evaluation is limited to standard continual learning models (*e.g.*, EWC, AFEC) and does not assess the attack's efficacy against models specifically designed for adversarial defense in continual learning settings. Recent work, such as the Sustainable Self-evolution Adversarial Training (SSEAT, MM '24) framework and "Continual Adversarial Defense" (Wang et al.), represents important baselines that should be included. Benchmarking AADB against these dedicated defenses is crucial to rigorously demonstrating its potency and properly contextualizing its contribution within the field.
- While the defense module shows some effectiveness, its performance against the most powerful "without error labels" attack remains limited (66.25% accuracy on adversarial samples in Table 5). The absence of comparative experiments with state-of-the-art adversarial defense methods (*e.g.*, those based on adversarial training) makes the contribution of the defense part less compelling.
- Although quantitative metrics are provided and a diffusion model is used for reconstruction, the main paper lacks visualized images of the reconstructed adversarial examples. This limits the reader's ability to assess visual stealthiness.
- The paper contains multiple grammatical errors (*e.g.*, missing articles, incorrect tenses on Lines 52, 111, 190) that should be corrected for clarity. Additionally, which baseline model is used, ResNet-16 (in Section 4.1) or  ResNet-18 (in Appendix A.2)?

**Questions:**

- How scalable is the current defense mechanism to larger-scale datasets (*e.g.*, ImageNet) or more complex tasks (*e.g.*, object detection)?
- Could you provide a computational cost analysis (*e.g.*, training time, GPU memory consumption) for the AADB framework (both attack and defense) on typical CL tasks? How much overhead does it add compared to baseline methods?

---

> ### Author Response · Authors · 2025-11-21
>
> Thank you for your suggestions. I will carefully consider them and make improvements.

---

### Note · Authors · 2026-01-12

I have read and agree with the venue's withdrawal policy on behalf of myself and my co-authors.